# Tumor Progression from a Fibroblast Activation Protein Perspective: Novel Diagnostic and Therapeutic Scenarios for Colorectal Cancer

**DOI:** 10.3390/diagnostics13203199

**Published:** 2023-10-13

**Authors:** Martina Rossetti, Stefano Stanca, Rossella Del Frate, Francesco Bartoli, Andrea Marciano, Enrica Esposito, Alessandra Fantoni, Anna Paola Erba, Piero Vincenzo Lippolis, Pinuccia Faviana

**Affiliations:** 1Department of Surgical, Medical, Molecular Pathology and Critical Area, University of Pisa, 56126 Pisa, Italy; martina.rossetti@outlook.com (M.R.); stefano.stancalucia@gmail.com (S.S.); rossella@malg.it (R.D.F.); alessandra.fantoni@icloud.com (A.F.); 2Department of Translational Research and New Technologies in Medicine and Surgery, University of Pisa, 56126 Pisa, Italy; francesco.bartoli@unipi.it (F.B.); andrea.marciano2305@gmail.com (A.M.); dr.enrica.esposito@gmail.com (E.E.); 3Department of Medicine and Surgery, University of Milan Bicocca and Nuclear Medicine Unit ASST Ospedale Papa Giovanni XXIII Bergamo, 24127 Bergamo, Italy; paolaanna.erba@unimib.it; 4Department of General Surgery, University Hospital of Pisa, 56124 Pisa, Italy; p.lippolis@ao-pisa.toscana.it

**Keywords:** colorectal cancer, tumor microenvironment, CAFs, FAP, imaging, radiometabolic therapy

## Abstract

In 2020, the Global Cancer Observatory estimated the incidence of colorectal cancer (CRC) at around 10.7% coupled with a mortality rate of 9.5%. The explanation for these values lies in the tumor microenvironment consisting of the extracellular matrix and cancer-associated fibroblasts (CAFs). Fibroblast activation protein (FAP) offers a promising target for cancer therapy since its functions contribute to tumor progression. Immunohistochemistry examination of FAP, fibronectin ED-B, and CXCR4 in primary tumors and their respective synchronous and/or metachronous metastases along with semiquantitative analysis have been carried out on histological samples of 50 patients diagnosed with metastatic CRC. The intensity of FAP, articulated by both “Intensity %” and “Intensity score”, is lower in the first metastasis compared to the primary tumor with a statistically significant correlation. No significant correlations have been observed regarding fibronectin ED-B and CXCR4. Tumors that produce FAP have an ambivalent relationship with this protein. At first, they exploit FAP, but later they reduce its expressiveness. Although our study has not directly included FAP-Inhibitor (FAPI) PET/CT, the considerable expression of FAP reveals its potential as a diagnostic and therapeutic tool worthy of further investigation. This dynamic relationship between cancer and FAP has substantial diagnostic and therapeutic implications.

## 1. Introduction

As a neoplastic entity, colorectal cancer (CRC) arises from a single cell escaping cellular control mechanisms [1], whose dysregulated proliferation results in a solid tumor. Surveys on CRC conducted by the International Agency for Research on Cancer (IARC) highlight its significant social impact with 1.9 million people diagnosed with CRC every year [2,3]. Over the past decades, knowledge about the epidemiology, pathophysiology, and molecular biology of CRC has considerably expanded. Yet, in 2020, the Global Cancer Observatory estimated the incidence of this cancer at around 10.7% coupled with a mortality rate of 9.5% [4] and predicted that in 2040 there would be 3.2 million new cases [2]. It is important to emphasize the increase in the incidence in <50–year–old people, especially because it is this cohort that will have a greater and long-term impact on public health [5,6,7]. Given this value, what are the factors at the basis of such a high level of mortality for a pathology widely studied by the scientific community? This question is significant even in light of the increase in new surgical techniques and targeted immunotherapies. It is our conviction, supported by specific studies in the literature [8,9,10,11], that the answer lies beyond the mere neoplastic cell, in the so-called tumor microenvironment (TME). As a consequence, this work has been conceived to address the tumor from the perspective of TME. The focus of this study aims at dissecting the stromal component of the TME, particularly the role and expression of the fibroblast activation protein (FAP), fibronectin ED-B, and CXCR4 in primary tumors and their respective metastases.

The TME consists of tumor-infiltrating cells, such as tumor-associated macrophages (TAMs), cancer-associated fibroblasts (CAFs), monocytes, neutrophils, natural killer cells, endothelial cells, and mesenchymal stem cells. Additionally, it includes the extracellular matrix (ECM), which is composed of collagen, laminins, fibronectin, proteoglycans, and hyaluronans [12,13,14]. These components sustain tumor-associated inflammation, angiogenesis, and tumor spreading [13]. The TME is a dynamic “ecosystem”. Initially, it does not represent a pro-tumor element, it is due to the continuous and dynamic interactions of its components that the support of the tumor becomes its primary objective [13]. It is a “niche” where neoplastic cells proliferate autonomously and establish two-way communication with everything surrounding them. The TME guarantees the survival of the neoplasm, both in terms of proliferative ability and in evading immune mechanisms triggered to suppress uncontrolled replication [15]. In fact, the TME influences tumor biology, responses to therapy, and clinical outcomes [16]. Furthermore, understanding the cross-talk between tumor cells and the TME is critical for the identification of new therapeutic targets.

The ECM is a three-dimensional structure that aims at maintaining tissue integrity, regulating cell migration and proliferation, and providing a pool of cytokines and growth factors [13]. The ECM includes collagen, glycoproteins, elastin, and proteoglycans [17]. In the TME, the ECM acquires a “disordered” conformation accumulating collagenous matrix and recruiting numerous fibroblasts in the framework of a process known as “epithelial–mesenchymal transition” (EMT). The EMT represents the very initial step of metastasis [18,19] and it is mainly supported by molecules such as TGF-β, TNF-α, and Metalloproteinases (MMPs) produced by TAMs and by the activation of the Smad pathway [13]. MMPs intensify the ECM disintegration [20], while the TGF-β/Smad signal, in fact, stimulates rapid cellular movement [21,22]. Thanks to the EMT mechanism, we observe the loss of epithelial cell features associated with the gain of mesenchymal characteristics [20]. Transformed epithelial cells can perform the functions of stromal cells with the aim of accelerating tumor growth and metastasis [13].

The ECM acquires the ability not only to promote neoplastic progression but also to shield and facilitate cancer spread [23]. Since the cells surround themselves with a dense collagenous stroma, they become more difficult to reach not only for immune system cells but also for the molecules induced by immunotherapeutic drugs. Moreover, this sort of barrier results in a hypoxic condition triggering tumor neo-angiogenesis mechanisms [17].

Like the ECM, fibroblasts also play a crucial role in the TME. Under physiological conditions, they are involved in the repair and formation of the extracellular stroma embodying the architectural basis to support the growth of the neoplasm. When fibroblasts are recruited by the tumor, they change conformation and stimulate the EMT through the production of TGF-β, thereby becoming active cells in maintaining the neoplasm. Neoplastic cells, in fact, are able to mobilize the precursors of fibroblasts and transform them into cancer-associated fibroblasts (CAFs) [24].

CAFs are the most representative cells of the TME; nevertheless, their abilities are still not fully clarified. These cells are identified on the basis of the expression of various molecules and one of them is FAP [13]. It has been noticed that CRC showing high levels of stromal FAP are more likely to be aggressive and to metastasize [25]. CAFs produce both growth factors and proteases and ECM constituents such as osteopontin and fibronectin that are able to promote tumor cell proliferation, survival, and migration [13]. In particular, fibronectin is a high-molecular-weight adhesive glycoprotein able to bind collagen, integrins, and fibrin, thus playing a fundamental role in cell relocation [13]. Intriguingly, some studies suggest that FAP may also be expressed by other cell types within the TME, such as macrophages, mature adipocytes, and mesenchymal stromal cells [26,27].

Given the pro-tumoral functions of CAFs, therapies directed against CAFs have been developing. In this context, FAP represents an ideal target for CAFs elimination [8]. Pre-clinical studies show that the use of a DNA vaccine that targets FAP efficiently reduces CAFs and subsequently contains CRC progression and metastasis [28].

These cells produce a range of signal molecules, including growth factors, cytokines, and chemokines such as CXCL12, CXCL14, CXCL16, CCL2, CCL5, IL-4, and IL-6, and metalloproteinases [29], which promote neoplastic progression.

In the following sections, we will delve deep into the methods, findings, and implications of this study, providing an analytical lens through which to evaluate the behavior of CRC in the context of the TME.

Other factors secreted by stromal cells include hepatocyte growth factor (HGF), fibroblast growth factors (FGF) 1 and 2, stromal cell-derived factor 1 (SDF1/CXCL12), and its associated receptor C-X-C chemokine receptor type 4 (CXCR4) [8,9]. The role of CXCR4 in cancer is critical, particularly in promoting cancer cell proliferation. Numerous studies have indicated a heightened expression of CXCR4 in neoplastic tissues, underscoring its potential value as a diagnostic biomarker [8,9]. CAFs also play a role in angiogenesis through the cleavage products of their substrates, as demonstrated by studies showing a correlation between FAP expression and micro-vessel density in tumors [10,11]. FAP stands out as a central protein within the framework of CAFs. This serine protease, predominantly found in CAFs and scarcely in normal tissues, offers a prospective target for cancer therapy [30]. Besides its connection with angiogenesis and tumor micro-vessel density [31], FAP has a role in regulating inflammation in the tumor milieu. Its functions may lead to immunosuppressive conditions, creating a favorable setting for tumor expansion and resistance. Numerous research efforts are focusing on FAP’s therapeutic potential due to its presence in CAFs [32]. Current investigations are exploring both antibodies and small molecules targeting FAP, with the intention to either inhibit its activities or utilize it as a means to convey the drug directly to the TME. This molecule induces an increase in the expression of mesenchymal markers, including vimentin and fibronectin [15,16]. This multifunctional glycoprotein promotes fibroblast migration, macrophage activity, cellular proliferation, adhesion, tumor cell migration, EMT, metastasis, and induction of an immunosuppressive environment [33]. High fibronectin expression is associated with cellular proliferation and poor prognosis [34].

The objectives of this study are twofold:To investigate the immunophenotypic expression of FAP, fibronectin ED-B, and CXCR4 in primary tumors of CRC and their liver, lung, and peritoneum synchronous and/or metachronous metastases.To preliminarily establish correlations between the tissue expression of the aforementioned stromal markers and the data potentially obtained from future PET imaging studies.

By focusing on these objectives, we aim to uncover new insights into the role of FAP and other stromal markers in CRC aggressiveness and their potential as therapeutic targets. This study serves as a groundwork for future research that could integrate immunohistochemical data with advanced imaging techniques like PET/CT for improved CRC management.

## 2. Materials and Methods

A retrospective analysis was conducted on histological samples of 50 patients diagnosed with metastasis of CRC who underwent surgical resection at the Surgery Division of the University Hospital of Pisa between 2012 and 2020. Of the 50 patients enrolled in the study, 20 were female and 30 were male. The average age of female patients is 66 years, while that of male patients is 62 years; resulting in an overall mean age of 63.6 years.

We analyzed the following cases:50 primary tumors;50 1st metastases;17 2nd metastases;7 3rd metastases;1 case of 4th metastasis.

Since 25 patients developed multiple metastases at different time stages of the disease, we indicated as “1st metastasis” those that, from the chronological point of view, developed and were diagnosed first.

For the majority of the cases (47 patients), molecular data regarding KRAS, NRAS, and BRAF mutations were available, while microsatellite status was evaluated in 42 samples. All patients (*n* = 50) had at least 1 liver metastasis: 16 of these (32%) had a metachronous liver metastasis and 34 (68%) a synchronous one. In total, 5 subjects also developed lung metastasis: 4 of these had a metachronous metastasis (80%), and only 1 experienced a synchronous pulmonary localization (20%). Lastly, 3 patients showed peritoneal metastasis, synchronous in all the cases.

Notably, pathologists use the tumor–node–metastasis (TNM) classification for the CRC staging [35]. The T parameter refers to the infiltration of the tumor in the diverse layers of the bowel wall (T1, submucosa; T2, muscularis propria; T3, mesocolic or mesorectal fat; T4a, perforation of the visceral peritoneum; and T4b, ingrowth in other organs). The N relates to how many lymph nodes are colonized by the neoplastic cells (N0, no involved lymph nodes; N1a, one regional lymph node involved; N1b, 2–3 lymph nodes involved; N2a, 4–6 lymph nodes; N2b, 7 or more). Finally, the M factor pertains to the presence of distant metastasis (M1a, spread to 1 other part of the body beyond the colon or rectum; M1b, spread to more than 1 part of the body other than the colon or rectum; and M1c, spread to the peritoneal surface). The three parameters are combined for the final staging [36] as shown in Figure 1.

Of our cases, 43 patients were classified as Stage IV given the manifestation of synchronous metastasis (30 Stage IVA, 2 IVB, and 2 IVC), while 16 cases showed as <IV Stage because of the presence only of metachronous metastasis (1 Stage I, 4 IIA, 1 IIC, 6 IIIB, and 4 IIIC).

### 2.1. Statement of Ethics

This study was conducted in accordance with the World Medical Association Declaration of Helsinki and approved by the Ethical Committee of the University of Pisa (Protocol# 9989 from 20 February 2019). Written informed consent was obtained from each patient.

### 2.2. Immunohistochemistry

Formalin-fixed, paraffin-embedded blocks were used for immunohistochemical staining. The expression of FAP, fibronectin ED-B, and CXCR4 was assessed on FFPE tumor tissue samples using immunohistochemistry (IHC). Anti-FAP, α- antibody (SP325 Abcam, prediluted antibody, for each section is a drop of preparation), anti-fibronectin (EP5 Santa Cruz, 1:50), anti-CXCR4 antibody (EPUMBR3 Abcam,1:100) were used. To obtain the immunoreaction, the avidin-biotin peroxidase method was assessed. IHC analysis was performed on the Ventana Medical System, with appropriate positive controls for all cases. Staining of known positive cases and omission of the first antibody were used as positive and negative controls, respectively.

### 2.3. Semiquantitative Analysis of Immunohistochemistry

The FAP slides were examined under a light microscope at 40× power by three independent observers (S.S., A.F., and P.F.) unaware of each other’s evaluation. Semiquantitative analysis of stromal staining was scored as 0, 1+, 2+, and 3+ (51–100% stromal staining) [21,22]. Grade 0 was defined as complete absence or weak FAP immunostaining in <1% of tumor stroma; grade 1+ was focal positivity in 1–10% of stromal cells; grade 2+ was FAP immunostaining positive in 11–50% of stromal cells; and grade 3+ was FAP immunostaining positive in >50% of stromal cells. A global assessment of the entire tumor was performed without selecting the invasive front or areas of active tumor growth. Further assessment of maximal staining intensity was performed and classified as none, weak, intermediate, or strong [37].

Regarding fibronectin, positive cells were quantified as a percentage of the total number of tumor cells and assigned to one of the following categories: 0, <5%; 1.5–24%; 2.25–49%; 3, 50–74%; and 4, ≥75%. The percentage of positive tumor cells and staining intensity were multiplied to generate an immunoreactivity score (IS) for each case. IS values ranged from 0 to 12; IS ≥ 3 was considered positive, whereas IS < 3 was negative. Stromal FN ED-B expression was graded into three categories: no or weak staining, no staining, or a low number of FN ED-B-positive strands; moderate staining, fine FN ED-B-positive strands; and strong staining, coarse FN ED-B-positive strands. The intensity of staining for CXCR4 was scored semi-quantitatively as follows: 1, weak; 2, medium; 3, strong; and 4, very strong. The percentage of maximally stained tumor/epithelial cells in each section was recorded (0, none; 1, <30%; 2, 30–50%; 3, >50%). Overexpression of cytoplasmic or nuclear CXCR4 was defined as a combined score for the intensity and area of staining that was >3 (nuclear expression of CXCR4 is associated with advanced CRC). Figure 2 summarizes the semiquantitative staining for FAP and CXCR4 in primary tumor and liver metastasis, respectively.

### 2.4. Aims

We investigated the following key points:(1)The correlation between the expression of fibronectin and CXCR4 in both the primary tumor and metastatic tissues in terms of intensity, number of positive cells, and intensity score.(2)The correlation between the expression of FAP, in terms of intensity % and intensity score, in the primary tumor and 1st metastasis.(3)The differences in terms of expression intensity of FAP and CXCR4 in the subgroups of patients with and without the mutations in BRAF, KRAS, and NRAS.(4)The correlations between BRAF, KRAS, NRAS mutations, and FAP and CXCR4 in both the primary tumor and the 1st metastasis.

## 3. Results

### Statistical Analysis

Data are presented as absolute and relative frequencies. Using the chi-square test, the expression of “Intensity %” and “Intensity score” in primary tumors and metastases were compared as well as in subgroups with BRAF, KRAS, and NRAS mutations. Spearman’s correlation coefficient and Kendall’s Tau test were utilized for correlation analyses to assess the relationship between the variables obtained from the primary tumor and the first metastasis. The Kruskal–Wallis test was used in the evaluation of FAP and CXCR4 in patients with and without BRAF, KRAS and NRAS mutations. All statistical analyses were performed by MedCalc software (MedCalc Statistical Software version 18.2.1, Ostend, Belgium; 2018). The variables taken into consideration are BRAF, KRAS, NRAS mutations, the expression of fibronectin, CXCR4, and FAP in terms of intensity in both the primary tumor and metastases, and the number of metastases. KRAS mutation did not show any association with the number of metastases in the same patient (see Table 1 for a summary of FAP statistical analyses).

(1)Due to the sample size of the investigated classes, the comparison focused on the primary tumor vs. the first metastasis. No statistically significant correlations emerged in comparing fibronectin with CXCR4 between the primary tumor and the first metastasis.(2)However, the evaluation of FAP in terms of “Intensity %” resulted in a Kendall’s Tau of 0.293 with a significance level of *p* = 0.0028, and FAP “Intensity score” revealed a Kendall’s Tau of 0.31 with a significance level of *p* = 0.0016. In both cases, a medium association was indicated.

Figure 3 and Figure 4 demonstrate that the intensity of FAP expression, articulated in both “Intensity %” and “Intensity score” appears to be lower in the first metastasis compared to the primary tumor. In 30 out of 50 pairs, FAP is reduced in the first metastasis compared to the primary tumor, while it remains stable in 13 out of 50 pairs and increases in 7 out of 50 pairs.

Regarding “Intensity score”, in 26 out of 50 pairs, the expression of FAP was reduced in the 1st metastasis compared to the primary tumor, while it remained stable in 19 out of 50 pairs and increased in 5 out of 50 pairs.

(3)The Kruskal–Wallis test was also performed to assess the difference in expression intensity of FAP and CXCR4 in the subgroups of patients with and without the mutations in BRAF, KRAS and NRAS. No statistical differences between the subgroups have been detected. Regarding FAP, although no statistically significant, higher values emerge in both the primary tumor and the 1st metastasis in patients with KRAS mutation respecting those without mutations, as shown in Figure 5 and Figure 6.

(4)Furthermore, the correlation between BRAF, KRAS, NRAS mutations, and FAP and CXCR4 in both the primary tumor and the first metastasis were examined. Due to the limited sample, the mutational assessment highlighted just the presence without specifying the mutation type. No statistically significant values emerged in associating CXCR4 or FAP expression with BRAF, KRAS, and NRAS mutations.

## 4. Discussion

Tumor metastasis is a multistep process that involves the escape of neoplastic cells from the primary site, their survival in circulation, and their seeding and proliferation at distant sites. Each of these stages incorporates rate-determining steps influenced by non-malignant cells in the TME [38]. CRC metastases are generally considered to be regulated by interactions between tumor cells and tumor-activated stromal factors, particularly with CAFs within the TME [39].

FAP is a marker of CAFs and its expression is higher in aggressive tumors [40] that are more likely to metastasize [41]. In this study, we analyzed FAP behavior in the stroma of both primary and metastatic CRS.

According to our results, while there is no significant correlation between fibronectin and CXCR4 in the primary tumor and the metastasis, FAP is reduced in the metastatic site as compared to the primary tumor. This result suggests that the relation between the primary neoplasm and FAP changes over time. As previously described in the Introduction, FAP is an essential element for tumor growth. The neoplasm takes advantage of this molecule produced by CAFs in the TME to evade the immune system, grow, and become more aggressive [42]. This is why several studies have shown that the more FAP is produced in TME, the more aggressive the tumor is, and the poorer the prognosis [43]. However, this relationship does not remain constant. The significance of the results obtained from this study shows that in the metastatic environment, neoplastic cells do not exploit FAP to survive and therefore the expression of this protein is reduced. The tumor takes advantage of FAP in a “static” phase of its growth. In the primary site, neoplastic cells have to grow, replicate, and defend themselves; they need to create a defensive “ecosystem”. As the tumor acquires the ability to spread, its behavior becomes more “dynamic”. In this context of mobilization, diffusion, and dynamism, it would be counterproductive for the tumor itself to continue to express high levels of FAP. This explains why, already in the first metastasis, FAP is significantly reduced compared to the primary tumor. Our results, therefore, highlight how CRC exploits in a different way, depending on its needs, the molecules that surround it.

KRAS mutation is considered to be a negative predictive factor [44] as it results in a lower response to anti-EGFR therapy [45].

Although the data obtained for Aims 3 and 4 of this study (see Materials and Methods section) are not statistically significant, they show a trend of higher FAP production in primary tumors compared to the first metastasis when KRAS mutation is observed.

This information appears to be in line with the tendency to consider KRAS mutated-CRC more aggressive than not-mutated CRC. Despite the fact that our results on KRAS mutated-CRC were not significant, the data show that, again, FAP is still associated with more aggressive tumors.

In light of this, it can be stated that the negative impact that FAP has on the survival of patients diagnosed with CRC is due to its stromal expression not at the metastatic site but precisely at the level of the primary tumor.

The significance of our results demonstrates the fundamental role played by FAP in neoplastic growth. Tumors synthesizing FAP have an ambivalent relationship with this protein. If, at the first stage, they exploit FAP, later they tend to reduce its expressiveness. This dynamic relationship between cancer and FAP has substantial diagnostic and therapeutic implications.

In this context, our preliminary analyses show elevated levels of FAP in the CRC TME, pinpointing a new focus for future diagnostic and therapeutic investigations. In the initial phases of the assessment of the disease, the visualization of the FAP through radio-diagnostic techniques allows a better perception of the three-dimensionality of the neoplasm and consequently allows a better staging. In the assessment of CRC, Fluorine-18 fluorodeoxyglucose (18F-FDG) PET/CT serves as an important imaging tool [46]. However, it has limitations. Due to the variations in glucose uptake by the different CRC histological histotypes, [^18^F]FDG PET/CT exhibits low sensitivity in detecting primary lesions [47]. This study has potential limitations. The obtained results are based on retrospective analyses of IHC and molecular data. Aware that, to date, the use of the aforementioned imaging techniques is not a routine step in patient management, this study aims at providing preliminary data that represent a starting point for the possible insertion of binding-FAP tracers in diagnostic practice. Although we have did not use molecular imaging techniques like FAP-Inhibitor (FAPI) PET/CT, the implications of our findings emphasize the need for in-depth research into molecular imaging modalities that target FAP. FAPI PET/CT specifically targets fibroblasts in the TME, enabling visualization of the ECM. This attribute may confer unique advantages in the morphological depiction of solid tumors.

Based on our findings, it is also possible to state that the use of FAP in tumor diagnostics is able to provide better estimates when employed to evaluate the not-metastasizing primary tumor. In this same phase of disease, FAP would acquire not only a diagnostic role but potentially also a therapeutic one. Once the FAP+ tumor is visualized, the development and use of an anti-FAP drug would lead to the attack of the fibrous “niche” that the tumor exploits to protect itself. FAP upregulation fosters a pro-inflammatory environment by inducing specific cytokines, thereby altering the recruitment or function of immune cells like macrophages and T lymphocytes [40]. The implications of such modulation are profound. Not only can FAP facilitate an immunosuppressive environment that promotes tumor growth and metastasis, but it can also cause the tumor to be more resistant to therapies [48]. This application would open a breach through the “defensive walls” of the neoplasm that would become more targetable by the antineoplastic drugs.

## 5. Conclusions

Our study highlights the significant FAP increase in CRC supporting its role as a promising target in tumor biology. FAP is expressed in both the primary CRC tumor and the first metastasis but it is significantly reduced in the metastatic site. This result can be interpreted as the ability of the tumor to take advantage of the TME components in different ways at different times.

Although our study has not directly included FAPI PET/CT, the considerable IHC expression of FAP reveals its potential as a diagnostic and therapeutic tool worthy of further investigation. Consequently, FAP has emerged as a promising molecular target for novel therapeutic strategies [49]. Given these attributes, one of our focuses was to scrutinize the role of FAP not only in diagnostic imaging but also as a potential therapeutic target. FAP-inhibitor radiopharmaceuticals have been designed for PET/CT imaging, demonstrating a favorable biodistribution in normal tissues, a high tumor-to-background ratio, and high efficacy of [68Ga]FAPI PET/CT in staging and restaging tumors eligible for neo-adjuvant treatment and surgery [50,51,52,53]. This imaging technique provides crucial information for early diagnosis, precise staging, and therapy guidance. This progress holds promise for improved diagnosis and treatment in the field of cancer therapeutics. Our analyses represent, therefore, a theoretical premise to deepen the role of FAP in translational studies that also involve nuclear medicine.

The significance of our results shows that FAP is crucial in tumor progression. If, at first, tumors exploit FAP, later they tend to reduce its production. This dynamic relationship between cancer and FAP has substantial diagnostic and therapeutic implications.

## Figures and Tables

**Figure 1 diagnostics-13-03199-f001:**
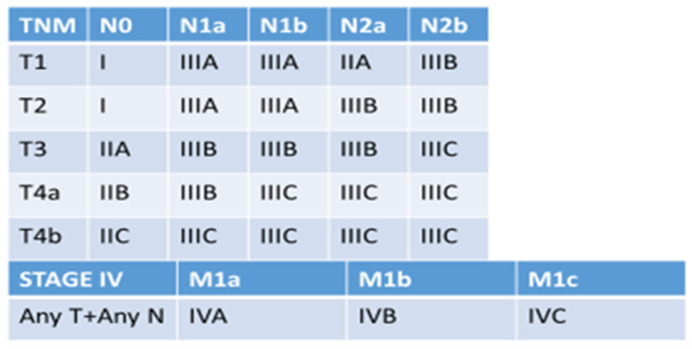
The CRC staging based on the TNM classification.

**Figure 2 diagnostics-13-03199-f002:**
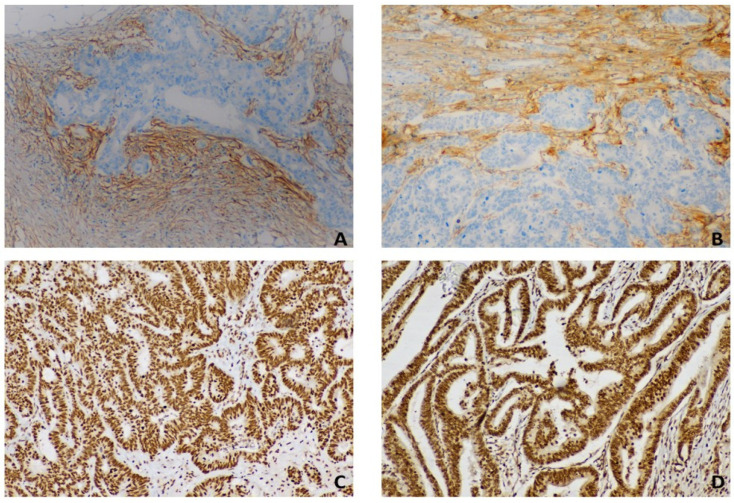
Cytoplasmic FAP staining involves the stromal CAFs next to the neoplastic cells. CRC primary site (**A**) and liver metastasis (**B**) show FAP-positive CAFs demarcating tumor glands (20× magnification). Nuclear CXCR4 is strong and diffuse in both the primary tumor (**C**) and in liver metastasis (**D**) (20× magnification).

**Figure 3 diagnostics-13-03199-f003:**
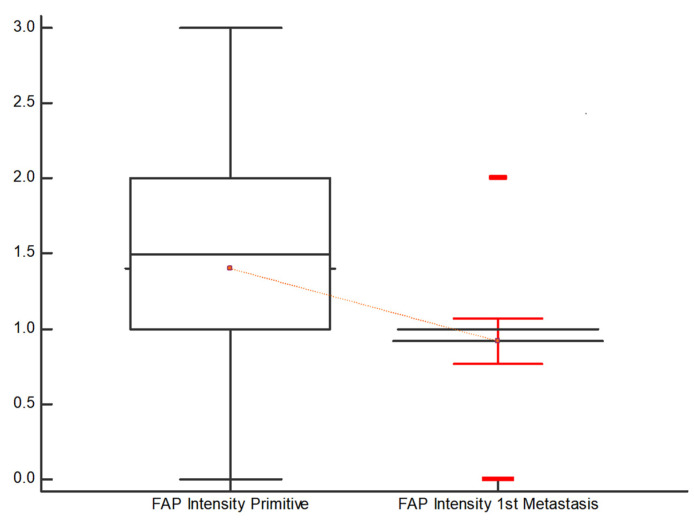
Distribution of the FAP expression based on the “Intensity score” in the primitive tumor and in the 1st metastasis. FAP is significantly lower in the 1st metastasis compared to the primary tumor.

**Figure 4 diagnostics-13-03199-f004:**
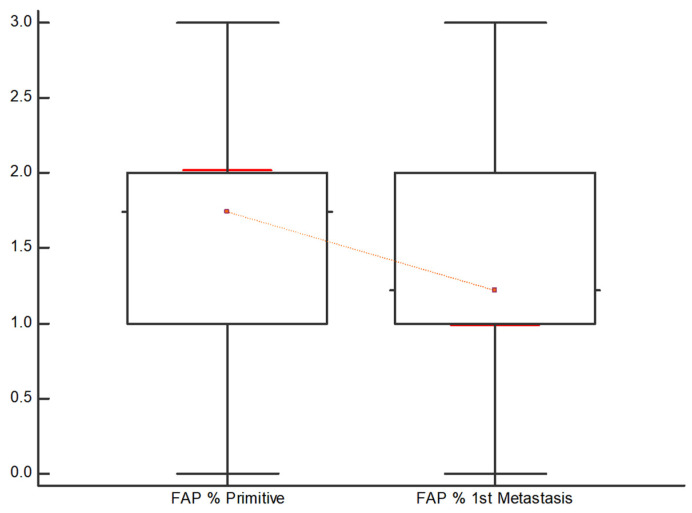
Distribution of the FAP expression based on the “Intensity %” in the primitive tumor and in the 1st metastasis. FAP is significantly lower in the 1st metastasis compared to the primary tumor.

**Figure 5 diagnostics-13-03199-f005:**
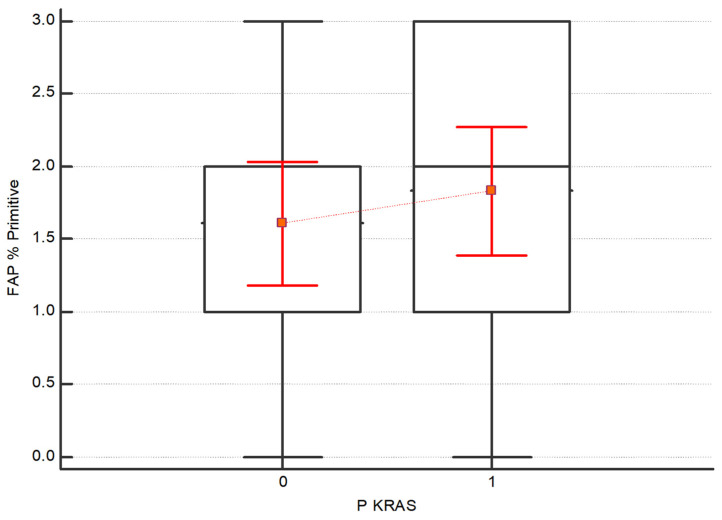
Distribution of FAP in the subgroups with and without KRAS mutation. FAP shows higher values in both the primary tumor and the 1st metastasis in patients with KRAS mutation compared to those without mutations; nevertheless, the results are not statistically significant (*p* value = 0.39).

**Figure 6 diagnostics-13-03199-f006:**
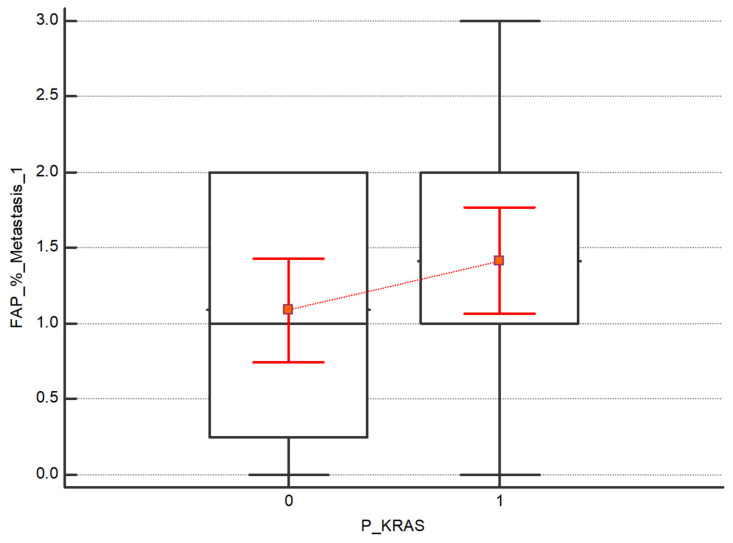
Distribution of FAP in 1st metastasis in the subgroups with and without KRAS mutation. FAP shows higher values in both the primary tumor and the 1st metastasis in patients with KRAS mutation compared to those without mutations; nevertheless, the results are not statistically significant (*p* value = 0.26).

**Table 1 diagnostics-13-03199-t001:** FAP expression as “Intensity %” and “Intensity score” in primary tumors and 1st metastases. The distribution is based on BRAF, KRAS, and NRAF mutations.

FAP % Primitive							FAP Intensity Primitive						
	0	1	2	3	Total (*n*)	*p* Value		0	1	2	3	Total (*n*)	*p* Value
**BRAF**						0.1165	**BRAF**						0.7038
Negative (%)	18.6	18.6	37.2	25.6	43		Negative (%)	18.6	32.6	41.9	7	43	
Positive (%)	0	0	100	0	4		Positive (%)	0	50	50	0	4	
**KRAS**						0.7297	**KRAS**						0.9577
Negative (%)	17.4	21.7	43.5	17.4	23		Negative (%)	17.4	34.8	43.5	4.3	23	
Positive (%)	16.7	12.5	41.7	29.2	24		Positive (%)	16.7	33.3	41.7	8.3	24	
**NRAS**						0.7127	**NRAS**						0.4202
Negative (%)	17.8	17.8	42.2	22.2	45		Negative (%)	17.8	35.6	40	6.7	45	
Positive (%)	0	0	50	50	2		Positive (%)	0	0	100	0	2	
**FAP % Metastasis 1**							**FAP Intensity Metastasis 1**						
	**0**	**1**	**2**	**3**	**Total (*n*)**	***p* Value**		**0**	**1**	**2**	**3**	**Total (*n*)**	***p* Value**
**BRAF**						0.6521	**BRAF**						0.4335
Negative (%)	18.5	46.5	27.9	7	43		Negative (%)	18.6	69.8	11.6	0	43	
Positive (%)	0	50	50	0	4		Positive (%)	0	100	0	0	4	
**KRAS**						0.119	**KRAS**						0.2657
Negative (%)	26.1	39.1	34.8	0	23		Negative (%)	26.1	65.2	8.7	0	23	
Positive (%)	8.3	54.2	25	12.5	24		Positive (%)	8.3	79.2	12.5	0	24	
**NRAS**						0.8543	**NRAS**						0.1702
Negative (%)	17.8	46.7	28.9	6.7	45		Negative (%)	17.8	73.3	8.9	0	45	
Positive (%)	0	50	50	0	2		Positive (%)	0	50	50	0	2	

## Data Availability

Not applicable.

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
