# Peer review of "Tumor Progression from a Fibroblast Activation Protein Perspective: Novel Diagnostic and Therapeutic Scenarios for Colorectal Cancer"

_diagnostics, 2023, doi:10.3390/diagnostics13203199_

Round 1

Reviewer 1 Report

In this study, immunohistochemistry examination of the expression of FAP, fibronectin ED-B, and CXCR4 in primary tumours and their respective synchronous and/or metachronous metastases and semiquantitative analysis have been carried out on histological samples of fifty patients diagnosed with metastasis of CRC. The intensity of FAP expression appears to be lower in the 1st metastasis compared to the primary tumour with a statistically significant correlation. This underscores the importance of therapeutic strategies targeting FAP. 

The aim is to establish correlation between tissue expression of stromal markers and data obtained from PET images, as well as the molecular and genomic features of the tumour. 

Although the point of focus seems to be good. However, this manuscript is considered to be inappropriate as a study because comparison with the intended PET image has not been carried out as written in the purpose.

No comments.

Author Response

Dear Reviewer 1,

thank you for your suggestions that allowed us to improve our study.

This study was conducted experimentally by immunohistochemistry and we duly recognize in this new version, as indicated by you, the absence of FAP PET/CT images.

We highlighted, thanks to your considerations, the limitations of our study that data obtained by our analyses aims at providing preliminary data that represent a starting point for the possible insertion of binding-FAP tracers in the diagnostic practice.

We thank you for your important contribution to this article and remain at your disposal for everything.

Sincerely yours,

Dr. Pinuccia Faviana

Reviewer 2 Report

The manuscript provides valuable insights into the incidence and tumor microenvironment of Colorectal cancer (CRC) and introduces the potential significance of Fibroblast activation protein (FAP)-positive fibroblasts in CRC therapy. The use of immunohistochemistry to examine FAP, fibronectin ED-B, and CXCR4 expression in primary tumors and metastases in a cohort of fifty CRC patients is a robust approach to investigating these factors. This manuscript presents a well-conducted study with significant implications for CRC therapy. The findings related to FAP expression warrant further exploration and discussion. Additionally, addressing the limitations of the study and providing more context on the clinical utility of FAP PET/CT would strengthen the manuscript. Overall, this research contributes valuable insights to the field of CRC treatment and warrants consideration for publication. However, there are areas where major improvements could be made.

Detailed comments and questions:

1.     The introduction lacks a clear statement of the specific research focus and objectives of the study. It is essential to explicitly state what the study aims to investigate or uncover in the context of CRC and the TME. Providing a concise research question or hypothesis would guide the reader and set the stage for the subsequent sections.

2.     While the introduction extensively discusses the general background and importance of CRC and the TME, it could benefit from a smoother transition to introduce the study itself. It should clearly indicate that the subsequent sections will delve into the study's methods, findings, and implications.

3.     The Materials and Methods section lacks essential demographic information about the patient cohort, such as age, gender, and other clinical characteristics. This information is crucial for understanding the composition of the patient group and its potential implications for the study findings.

4.     While Figure 2 is referenced in the text, the figure legends should be more informative. They should provide context for each image, explaining what is depicted in each image, such as the specific staining observed and the magnification used.

5.     One of the primary issues with the results is the lack of interpretation of the results. While the statistical tests are mentioned, there is minimal discussion of what these findings mean in the context of colorectal cancer (CRC). The authors should provide explanations and interpretations of the statistical results to help readers understand their clinical and scientific significance.

6.     The presentation of data in the results could be improved. Data tables or summary statistics could be included to provide a clearer overview of the results, making it easier for readers to understand the key findings.

7.     The discussion could benefit from clearer organization. It covers several important aspects, including the role of FAP and CXCR4, the clinical implications of their expression, and the potential of FAP PET/CT. To enhance readability, consider breaking down the discussion into subsections or paragraphs with clear headings.

8.     While the discussion touches on the clinical implications of FAP and CXCR4 expression, it could be expanded to provide more concrete examples of how targeting these molecules may impact patient care. Discussing specific clinical scenarios or potential treatment strategies would make the clinical relevance more tangible to readers.

Minor editing of the English language required

Author Response

Dear Reviewer 2,

a sincere thank you for your comments and suggestions absolutely useful to improve our work. Following your comments, we worked with the utmost seriousness on the following points, as brought to light by you:

1-2) We expanded the background highlighting the importance of TME in CRC. Moreover, we edited, thanks to your indication, the concordance between the Introduction and the other Sections explaining the scientific context that brought to the conceptualization of out hypotheses and to the analyses that we have performed. This leads to a smoother transition and to a better reading.

3) We integrated the “Materials and Methods” section with the demographic information about the patient cohort,  such as age and gender, to meliorate the understanding of the composition of the patients group.

4) Figure 2 legend has been improved; we explained in detail the context of each image and added the information about the magnification that has been used.

5) We provided, as required, in the Discussion a comprehensive explanation of the obtained results along with their biological interpretation. Thanks to your suggestion, the Discussion now exhibits the scientific significance and implications of our results.

6) A table summarizing the statistical analyses has been inserted (see “Table 1”) to provide a clearer overview of the results.  

7-8) We edited, thanks to your indication, the organization of the Discussion by dividing the various paragraphs. Furthermore, we expanded this Section by discussing how targeting FAP may impact on patient care, the potential treatment strategies and the limitations of our study.

English Language has been object of accurate editing.

We thank you for your important contribution to this article. We hope having met your requirements and remain at your complete disposal for everything.

Yours sincerely,

Dr. Pinuccia Faviana

Reviewer 3 Report

In this manuscript, the author conducted a retrospective analysis of tissue samples from patients with colon cancer, assessing various markers, including FAP. The study is well-designed and presents data with sufficient detail. However, the discussion falls short in terms of elaborating on and interpreting the results. Given the current landscape in the field of FAPI imaging, the presented findings, indicating lower FAP expression in the first metastasis lesion, could hold significant value. However, the author did not delve into explaining how this difference might impact the application of FAPI or related FAPI-based agents. 

Author Response

Dear Reviewer 3,

thank you for your suggestions that allowed us to improve our study.

A scrupulous review was performed in order to improve the general coherence of the manuscript, focusing particularly, as required on the experimental phase, in particular, through the deepening in the Discussion about the potential application of FAP PET/CT in CRC assessment and treatment.

We hope having embraced your indications and satisfied your requirements. Clearly, we remain at your complete disposal for everything. A sincere thank you again.

Yours sincerely,

Dr. Pinuccia Faviana

Round 2

Reviewer 1 Report

1) Please explain the definition of 1st metastases.

2) FAP PET should be rephrased as FAPI (FAP inhibitor) PET.

N.A.

Author Response

Dear Reviewer 1,

thank you for your suggestions that helped us to improve our work. Following your comments, we worked on the points brought to light by you:

  • We explained, thanks to your considerations, the definition of the 1st metastases (see lines 509-511).
  • The locution “FAP PET” has been be rephrased as “FAPI (FAP inhibitor) PET”.

We thank you for your important contribution to this article and remain at your disposal for everything.

Sincerely yours,

Dr. Pinuccia Faviana

Reviewer 2 Report

The authors adressed all my comments and the manuscript in the present version.

It is ok.
